# CAN LLMS DESIGN REAL HARDWARE? A NEW BENCHMARK FOR RTL DESIGN AND VERIFICATION TASKS

## ABSTRACT

We present the XYZ benchmark [note to reviewers: name withheld in accordance with ICLR double-blind policy], a new dataset and infrastructure to advance LLM and agent research in hardware design and verification. XYZ includes 783 problems across 13 task categories, covering RTL generation, verification, debugging, specification alignment, and technical Q&A authored by experienced hardware engineers. Problems are offered in both non-agentic and agentic formats. The benchmark introduces more realistic and challenging contexts than prior work, with state-of-the-art models achieving no more than 34% pass@1 on code generation. Agentic tasks—especially those involving RTL reuse and verification—are particularly difficult. Evaluation uses open-source tools and model scoring infrastructure, with comprehension tasks assessed via BLEU and LLM-based judging. XYZ reveals substantial gaps in current model capabilities, underscoring the need for continued research toward robust, real-world hardware design automation.

## 1 INTRODUCTION

Large language models (LLMs) have seen widespread adoption in software development for code generation, bug fixing, question answering, test generation, and related tasks. Recently, agentic assistants such as Cursor (Anysphere Inc. (2025))—an AI-powered IDE based on Visual Studio Code—have gained traction for their ability to not only answer questions but also perform complex code edits and execute commands.

By contrast, semiconductor hardware design has not benefited as significantly from LLMs. Generating Verilog RTL (Register-Transfer Level—the textual code used to design digital logic chips) with LLMs presents unique challenges, including the limited availability of high-quality training data (Wang et al. (2025); Liu et al. (2025a)) and the relative recency of domain-specific benchmarks. Two widely used datasets are VerilogEval (Liu et al. (2023); Ho et al. (2025)) and RTLLM (Lu et al. (2024); Liu et al. (2025b)), which report pass rates as high as 63% on GPT-4 and 94% for agentic approaches (Pinckney et al. (2025); Ho et al. (2025)). However, these benchmarks are narrow in scope and do not reflect the full complexity of hardware development workflows. Moreover, their high pass rates leave little headroom for measuring future improvements, limiting their usefulness as research drivers.

VerilogEval and RTLLM rely on hand-crafted prompts and evaluate on small, self-contained problems. RTL-Repo (Allam and Shalan (2024)) introduces more realistic GitHub-derived contexts, prompting LLMs to complete redacted code regions. While it captures real-world structure, RTL-Repo focuses solely on code completion and does not test broader challenges like specification-to-RTL generation, debugging, or verification. Related benchmarks cover testbench stimuli (Zhang et al. (2025)), though close to 100% coverage of their benchmark is achievable by Claude 3.5 Sonnet, and formal assertions (Liu et al. (2025b)).

We introduce the *XYZ* benchmark [note to reviewers: name withheld in accordance with ICLR double-blind policy], which expands on prior work with broader task coverage and greater depth. XYZ includes 783 human-authored problems across 13 categories, including RTL generation, design verification, debugging, assertion creation, and technical comprehension. Tasks are provided in both Non-Agentic (single-turn) and Agentic (multi-turn, tool-using) formats. Previous benchmarks focus

on single-turn prompts and evaluation infrastructure, while XYZ is designed to evaluate agents, with support for tool interaction, iterative workflows, and complex reasoning.

XYZ addresses the growing need for benchmarks that reflect real-world hardware development. Problem categories cover tasks such as RTL/testbench generation, debugging, assertions, code modification, power and area optimization, question answering, and code-spec alignment. The dataset is intended to expand over time, evolving alongside improvements in LLM and agent capabilities, while continuing to offer meaningful challenge and headroom for future research.

This work makes four key contributions:

1. **The first agentic-oriented benchmark** for Verilog RTL code generation, verification, and related tasks. The benchmark's prompts and infrastructure are designed to evaluate Dockerized LLM-based agents on real-world problems with EDA tool use.

2. **A broader benchmark** that encompasses a wider range of hardware design and verification tasks. The benchmark is intended to support both model and agent research. Initial Non-Agentic categories were selected with greater agent workflows in mind, representing useful subtasks within larger design processes.

3. **A more challenging benchmark**, featuring tasks significantly more difficult than those in Verilo-gEval (Liu et al. (2023); Pinckney et al. (2025)) and RTLLM (Lu et al. (2024)). Prior benchmarks largely drew from public repositories and are increasingly saturated, with high pass rates from both models and agents. In contrast, the current benchmark offers data points crafted and QA'ed by experienced hardware engineers with more than 4 years of experience from scratch. As a result, we show that state-of-the-art models—including Claude 3.7 Sonnet, GPT-4.1, and LLaMA 3.1 405B—achieve no more than a 34% pass rate on code generation questions in our benchmark, providing substantial headroom for future research in LLM-driven hardware design.

4. **Analysis of model failures** examines why state-of-the-art models frequently fail across specific categories and offers insights into the key capabilities LLMs must develop before they can be reliably deployed for real-world hardware design and verification.

RTL code represents only a small fraction of public GitHub repositories compared to software code, and much design knowledge remains proprietary within industry. Consequently, there is a strong need for an advanced, human-written, publicly available benchmark dataset—composed of real-world design problems authored by design and verification experts. We created XYZ to address this critical gap.

## 2 XYZ DATASET

The XYZ dataset and infrastructure build on methodologies from software LLM benchmarks such as SWE-bench (Jimenez et al. (2024)) and Microsoft's Copilot evaluation harness (Agarwal et al. (2024)). Whereas SWE-bench had access to a wide range of high-quality, open-source, software code repositories and well-documented resolved GitHub issues to pull from, similar high-quality RTL repositories are not as available in the open-source domain. Instead, we engaged a team of approximately 35 hardware engineers with more than 4 years of Verilog and verification experience to author problems across 13 task categories and difficulty levels, in both *Non-Agentic* and *Agentic* formats.

In addition, subject matter experts with doctoral degrees in hardware design and/or engineering management experiences also reviewed each problem for accuracy, task fit, and appropriate scope, with intensive manual review during initial *calibration* batches to ensure data quality and task alignment. Once categories stabilized, LLM-based filtering was used to catch errors, such as missing context or incorrect category, and score ambiguity and consistency of the prompt. Sanity checks ensured all reference solutions passed and incomplete contexts failed as expected. Of the 1,313 problems written, 783 were retained after quality filtering described in Section 3. As with any codebase, a benchmark cannot be entirely bug-free (Ho et al. (2025)). Errors may cap maximum achievable scores, and updated benchmark versions will be released as needed.

Each datapoint, or "problem," represents a multi-file repository extracted at evaluation time. A test harness—typically a CocoTB (CocoTB (2025)) simulation script—assesses correctness based on

task type. CocoTB is a Python verification framework for testing RTL, and helps to automate the test harness. BLEU (Papineni et al. (2002)) scoring is used where code or natural language snippets are expected verbatim, while technical natural language answers are scored using LLM-based subjective judging.

We distinguish between the *testbench* (SystemVerilog provided in-context) and the *test harness* (used only for evaluation). Models or agents may generate or use a testbench but never see the test harness or reference solution.

## 2.1 TASK CATEGORIES

Categories in the initial XYZ release (Table 1) are grouped into two main areas: *Code Generation* and *Code Comprehension*. Code Generation covers RTL-focused tasks such as code completion, transforming natural language specifications to RTL, modifying or reusing existing modules, and improving code for linting or quality-of-results (QoR). It also includes design verification tasks like testbench stimulus and checker generation, assertion creation, and debugging. Code Comprehension includes matching specifications to RTL or testbench code (and vice versa), as well as technical question answering on both RTL and testbench content. These categories reflect common subtasks in real-world hardware design and verification workflows.

*Non-Agentic* problems are evaluated in a single-turn setting where the prompt and context are fully provided to the model. In contrast, *Agentic* problems run inside a Docker container, allowing an agent to inspect a mini-repository and invoke tools (e.g., simulators). For both Non-Agentic and Agentic problems we limited datapoint creation to *oracle contexts*, where models are provided only the minimal, relevant information needed to complete the task, bypassing the need for retrieval or broader context understanding. However, this is not a technical limitation of the benchmark infrastructure and a full-repository context could be added to future datapoints.

Category volumes were based on likely deployment scenarios. Most task categories include both Non-Agentic and Agentic datapoints, but some were designed as Non-Agentic-only or Agentic-only based on their expected use case—e.g., simpler tasks for single-turn model inference, and more complex tasks requiring tool use for agentic evaluation.

Each datapoint includes the context and a *golden* reference solution. Supporting materials—such as related module documentation, testbenches, or editable starter code—were included as needed. The benchmark is packaged as two JSONL files: one for Non-Agentic and one for Agentic datapoints. The table shows the mean and maximum prompt and context token counts for each category, as estimated using the `tiktoken cl100k_base` encoding.

| ID | Category Description | Volume | | Tokens Mean/Max (k) | |
|---|---|---|---|---|---|
| | | Non-Agentic | Agentic | Non-Agentic | Agentic |
| **Code Generation** | | | | | |
| **cid02** | RTL - Code Completion | 94 | 0 | 1.5/4.5 | – |
| **cid03** | RTL - Natural Language Spec to Code | 78 | 37 | 1.2/6.9 | 2.7/7.9 |
| **cid04** | RTL - Code Modification | 56 | 26 | 2.0/4.6 | 5.7/19.5 |
| **cid05** | RTL - Spec to RTL (Module Reuse) | 0 | 26 | – | 7.4/28.5 |
| **cid07** | RTL - Code Improvement (Linting/QoR) | 41 | 0 | 1.9/5.9 | – |
| **cid12** | Design Verification - Testbench Stimulus Gen. | 68 | 18 | 1.4/6.2 | 2.1/4.6 |
| **cid13** | Design Verification - Testbench Checker Gen. | 53 | 18 | 2.8/7.3 | 4.5/10.7 |
| **cid14** | Design Verification - Assertion Generation | 68 | 30 | 2.6/7.5 | 4.8/14.6 |
| **cid16** | Design Verification - Debugging / Bug Fixing | 36 | 11 | 2.3/6.5 | 3.9/14.5 |
| **Code Comprehension** | | | | | |
| **cid06** | Correspondence - RTL to/from Specification | 34 | 0 | 1.6/5.5 | – |
| **cid08** | Correspondence - Testbench to/from Test Plan | 29 | 0 | 3.1/6.1 | – |
| **cid09** | Question & Answer - RTL | 34 | 0 | 1.1/5.0 | – |
| **cid10** | Question & Answer - Testbench | 26 | 0 | 3.6/4.8 | – |
| | **Total # of Problems** | **617** | **166** | | |

Table 1: Comparison of Non-Agentic and Agentic problem counts by task category.

## 2.2 DATAPOINT AUTHOR GUIDELINES

Datapoint writers were instructed to cover a range of human-tagged difficulty levels—easy, medium, and hard. Since proxies like lines of code or gate count poorly capture true complexity (e.g., a 32-bit 16:1 multiplexer may be written succinctly or verbosely), writers were told to prioritize clarity and best coding practices over artificial complexity.

Non-Agentic problems include only easy and medium tasks, while Agentic problems span all difficulty levels, as hard problems are too complex for single-turn evaluation. Writers were also asked to diversify topical coverage within each category, including: (1) FSM and control logic (e.g., Mealy/Moore, arbitration, counters); (2) Arithmetic and datapath (e.g., adders, multipliers, shifters); (3) Interconnects (e.g., crossbars, routers, FIFOs); (4) Memory systems (e.g., caches, CAMs); and (5) Architecture (e.g., CPUs, accelerators).

## 3 BENCHMARK INFRASTRUCTURE

The benchmark infrastructure is implemented in Python and includes callback interfaces to evaluate custom models or agents. An overview of the evaluation flow is shown in Figure 1. Each datapoint can be run with either the initial context or the reference solution, enabling self-checking of harness validity. Harnesses use open-source tools where possible, including Icarus Verilog simulation (Williams (2025)), Yosys logic synthesis (Wolf and the YosysHQ contributors (2025)), and Verilator linting (Snyder and Contributors (2025)). Some tasks (cid12–14) require commercial tools, currently Cadence Xcelium (Cadence Design Systems, Inc. (2025). All agents and harnesses run inside Docker containers to isolate evaluation artifacts, ensure tool consistency, and maintain security. Users populate tool and agent images using provided templates. Configurable timeouts and retry counts accommodate varying compute access.

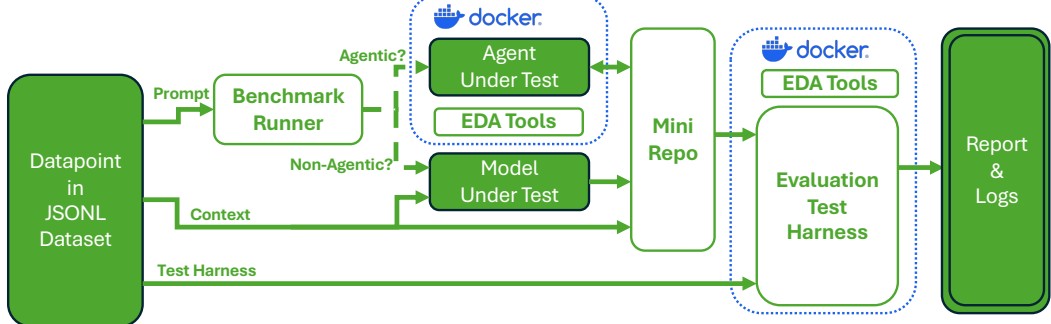

Figure 1: Benchmark Evaluation Flow.

The infrastructure includes a *map* feature for querying models across datapoints with custom prompts—useful for prompt refinement or batch evaluation. The map feature also supports automated quality filtering using an LLM judge to score datapoints and remove low-quality examples. Lastly, Agentic and Non-Agentic formats can be converted between to allow single-turn evaluation on Agentic problems or multi-turn agent evaluation on Non-Agentic problems.

## 4 LLM BENCHMARK RESULTS

We evaluated state-of-the-art models on the XYZ dataset, including both Non-Agentic and Agentic problems. Models evaluated include Anthropic Claude 3.7 Sonnet with and without Extended Thinking (Anthropic (2025)), Claude 3.5 Haiku, OpenAI GPT 4.1 (OpenAI (2025a)), GPT o1 (OpenAI (2024)), o4-mini OpenAI (2025b), Meta Llama 3.1 405B (Meta AI (2024a)), and Llama 3.1 70B (Meta AI (2024b)). We report a pass@1 with $n = 5$ samples as the pass rate. The pass@$k$ metric is the probability that at least one sample passes among $k$ samples, we estimate the expected value of pass@1 across $n = 5$ samples. For Llama 3.1 405B and 70B, we set the decoding parameters to

$T = 0.2$ and top-$p = 0.7$. For the other models we used the default temperature and top-$p$ supported by the API endpoint.

Tables 2 and 3 provide pass rates for the code generation tasks across models. Prior Verilog code generation benchmarks, such as VerilogEval v2 (Pinckney et al. (2025)), reported that LLaMA 3.1 405B achieved a pass rate of 57% on specification-to-RTL tasks, with GPT-4o achieving a pass@1 of 63%, the best result in that benchmark.

In contrast, the tables shows that XYZ presents a substantially greater challenge to state-of-the-art models. The highest aggregate pass@1 rate observed was 34% (Claude 3.7 Sonnet), followed by GPT-4.1—the successor to GPT-4o—at 29%, and LLaMA 3.1 405B at 23%.

Agentic problems, when evaluated in single-turn format using a model, were even more challenging overall—particularly for the OpenAI models. GPT-4.1 achieved a 21% pass@1 on Agentic tasks, 8% lower than its Non-Agentic score. Claude 3.7 Sonnet's pass rate dropped by 4% between Non-Agentic and Agentic problems, while LLaMA 3.1 405B showed only a 2% drop, likely reflecting its inability to solve many of the harder problems in either setting.

All reported results reflect the filtered dataset after automated quality control, as described in Section 2. Prior to filtering, pass rates were lower by approximately 3% and 1.5% on average for Non-Agentic and Agentic problems, respectively. These results highlight the difficulty of the XYZ benchmark and the significant advancements still required before LLMs can be reliably deployed in complex, real-world hardware design and verification workflows.

Generation pass rates vary significantly across categories, as shown in Table 2. Categories cid02–04 correspond to RTL code generation and modification, cid07 covers code improvement tasks (e.g., linting and QoR-focused modifications), and cid12–14 correspond to design verification tasks. Category cid16 is also included in the generation evaluation.

Design verification categories—specifically testbench stimulus and checker generation (cid12–13) and assertion generation (cid14)—exhibit substantially lower pass rates compared to other code generation categories. This is examined in more detail in Section 5. Notably, state-of-the-art LLMs consistently struggle to generate even syntactically valid testbench code, despite it being written in the same hardware description language (SystemVerilog) as the RTL code generation tasks. This discrepancy may stem from the more procedural and imperative nature of testbench code, as opposed to the declarative structure typical of RTL logic.

| Model | Overall | cid02 | cid03 | cid04 | cid07 | cid12 | cid13 | cid14 | cid16 |
|---|---|---|---|---|---|---|---|---|---|
| Claude 3.7 Sonnet | **33.56%** | 34.0% | 48.0% | 45.0% | 44.0% | 25.0% | 6.0% | 19.0% | 53.0% |
| " Thinking | **33.04%** | 35.0% | 44.0% | 44.0% | 45.0% | 24.0% | 7.0% | 23.0% | 51.0% |
| Claude 3.5 Haiku | **23.93%** | 28.0% | 40.0% | 32.0% | 28.0% | 16.0% | 3.0% | 11.0% | 31.0% |
| GPT 4.1 | **28.91%** | 37.0% | 44.0% | 37.0% | 32.0% | 16.0% | 10.0% | 12.0% | 45.0% |
| GPT o1 | **20.12%** | 20.0% | 31.0% | 30.0% | 23.0% | 15.0% | 9.0% | 5.0% | 33.0% |
| GPT o4-mini | **28.74%** | 35.0% | 47.0% | 44.0% | 27.0% | 13.0% | 11.0% | 10.0% | 43.0% |
| Llama 3.1 405B | **22.79%** | 24.0% | 31.0% | 36.0% | 20.0% | 21.0% | 5.0% | 13.0% | 32.0% |
| Llama 3.1 70B | **17.53%** | 18.0% | 20.0% | 33.0% | 21.0% | 16.0% | 4.0% | 7.0% | 26.0% |

Table 2: Non-Agentic Code Generation Problems: Pass Rates Across Categories and Models. Categories are grouped into RTL generation and modification, code improvement, testbench or assertion generation, and debugging. Results are reported as pass@1 with $n = 5$ samples.

Agentic datapoints were converted to Non-Agentic format for evaluation, as no open-source, general-purpose hardware design agent currently exists. Agentic generation pass@1 rates across categories, shown in Table 3, follow similar trends to those observed in Table 2. Code Completion (cid02) and Code Improvement (cid07) tasks are exclusive to the Non-Agentic dataset, while the Agentic dataset introduces Spec-to-RTL Module Reuse tasks (cid05). These problems require composing multiple existing RTL modules into a new top-level module, often with additional glue logic, to satisfy the specified behavioral requirements.

As in the Non-Agentic results, Claude 3.7 Sonnet performs notably well compared to other models on most RTL code generation and debugging categories (cid03–04, cid16). However, Claude 3.7 Sonnet does not exhibit a significant advantage over other models on Spec-to-RTL Component Reuse (cid05),

suggesting that while it excels at generating or modifying RTL code, it struggles with the more complex task of composing existing RTL components to implement new functionality.

| Model | Overall | cid03 | cid04 | cid05 | cid12 | cid13 | cid14 | cid16 |
|---|---|---|---|---|---|---|---|---|
| Claude 3.7 Sonnet | 29.0% | 49.0% | 42.0% | 24.0% | 7.0% | 0.0% | 19.0% | 53.0% |
| " Thinking | 29.0% | 39.0% | 44.0% | 24.0% | 7.0% | 1.0% | 28.0% | 56.0% |
| Claude 3.5 Haiku | 20.0% | 31.0% | 24.0% | 21.0% | 2.0% | 2.0% | 10.0% | 55.0% |
| GPT 4.1 | 21.0% | 31.0% | 24.0% | 21.0% | 4.0% | 13.0% | 13.0% | 45.0% |
| GPT o1 | 14.0% | 22.0% | 8.0% | 18.0% | 8.0% | 3.0% | 10.0% | 36.0% |
| GPT o4-mini | 20.0% | 32.0% | 16.0% | 20.0% | 6.0% | 7.0% | 16.0% | 38.0% |
| Llama 3.1 405B | 21.0% | 30.0% | 23.0% | 25.0% | 8.0% | 6.0% | 14.0% | 45.0% |
| Llama 3.1 70B | 15.0% | 23.0% | 13.0% | 18.0% | 6.0% | 6.0% | 6.0% | 45.0% |

Table 3: Agentic Code Generation Problems: Pass Rates Across Categories and Models. Categories are grouped into RTL generation and modification, testbench or assertion generation, and debugging. Results are reported as pass@1 with $n = 5$ samples.

The Code Comprehension dataset is limited to Non-Agentic format and is scored differently from the Code Generation problems. RTL/Testbench Correspondence tasks (cid06, cid08) are evaluated using BLEU (Papineni et al. (2002)) scores, as the expected responses are code or natural language snippets that should match a reference verbatim. RTL/Testbench Question & Answer tasks (cid09–10) are scored using subjective, LLM-based evaluation: the model compares an actual response against the reference solution in the context of the original prompt. The scoring prompt instructs the model to emphasize information explicitly requested in the original question. For efficiency and availability, GPT o4-mini is used as the scoring model.

As shown in the results, all LLMs perform well on the Question & Answer tasks, with minimal gains observed from newer models over older ones. Since conversational QA has been a central application area for LLMs, this may reflect the models' maturity in chatbot-style environments. However, further investigation is needed to assess the technical reliability of these scores.

| Model | Average Rating | cid06 | cid08 | cid09 | cid10 |
|---|---|---|---|---|---|
| Claude 3.7 Sonnet | 66.0% | 63.0% | 42.0% | 78.0% | 82.0% |
| " Thinking | 71.0% | 70.0% | 48.0% | 83.0% | 84.0% |
| Claude 3.5 Haiku | 51.0% | 25.0% | 27.0% | 73.0% | 83.0% |
| GPT 4.1 | 47.0% | 10.0% | 10.0% | 82.0% | 89.0% |
| GPT o1 | 43.0% | 8.0% | 1.0% | 82.0% | 83.0% |
| GPT o4-mini | 49.0% | 8.0% | 14.0% | 88.0% | 89.0% |
| Llama 3.1 405B | 40.0% | 10.0% | 1.0% | 75.0% | 78.0% |
| Llama 3.1 70B | 38.0% | 8.0% | 1.0% | 68.0% | 77.0% |

Table 4: Non-Agentic Code Comprehension Problems: Overall and Per-Category Scores. Categories are grouped into Correspondence and Question & Answer problems. Results are reported with $n = 5$ samples.

## 5 FAILURE ANALYSIS AND INSIGHTS

We perform a systematic and detailed category-level analysis of the failed cases for each LLM to identify the critical areas that need improvement in state-of-the-art LLMs across various Verilog design categories (i.e., RTL coding, assertion generation, testbench generation, debugging, etc.).

The category-level failure analysis flow is shown in Algorithm 1. First, we leverage a reasoning LLM (i.e., o1) to reflect on the failed data points and project the failure reflections into a vector space using SentenceTransformer (Line 2 to 5). Then, we apply the unsupervised K-means clustering methodology (Sinaga and Yang (2020)) to generate the optimal number of clusters based on the maximum silhouette score (Line 8 to 14). Finally, we use a reasoning LLM (i.e., o1) to interpret and summarize the category-level failures (CF), identifying the critical shortcomings of state-of-the-art LLMs in Verilog design and verification tasks (Line 15 to 18).

We present category-level failure analysis results for Llama 3.1 405B, Claude 3.7 Sonnet, and GPT 4.1 in Table 5. We report the number of failed cases, number of clusters, the failure entity of the

---

**Algorithm 1** Category-Level Failure Analysis

---

**Require:** Dataset $F_c = \{f_{c,1}, f_{c,2}, \ldots, f_{c,n}\}$,
  1: Set $F_e = []$                                                       {Failed reason embeddings}
  2: **for** each failed data point $f_{c,i} \in F_c$ **do**
  3:    $r_{c,i} = Reflect(f_{c,i})$                                         {LLM-based failure reflection}
  4:    $F_e.append(Embedding(r_{c,i}))$
  5: **end for**
  6: Set $s_{best} = 0$; $k_{best} = 0$
  7: Set $L_{best} = \text{zeros}(n, 1)$
  8: **for** $k \leftarrow 2$ to $11$ **do**
  9:    $L_k = Kmeans(F_e, k)$                                           {Kmeam clustering}
  10:    $s_k = silhouette\_score(F_e, L_k)$
  11:    **if** $s_k > s_{best}$ **then**
  12:       $s_{best} = s_k$; $k_{best} = k$; $L_{best} = L_k$;
  13:    **end if**
  14: **end for**
  15: **for** $g \leftarrow 0$ to $k_{best}$ **do**
  16:    $CF_g = Reason(F_{g,e})$            {LLM-based Category-Level Reasoning of cluster $g$}
  17: **end for**
  18: Return $CF$

---

largest cluster, and its percentage share of the total failed cases within each category. We observe that state-of-the-art LLMs particularly struggle with testbench stimulus generation (cid12), testbench checker generation (cid13), and assertion generation (cid14). Compared to RTL coding (cid02–cid04, cid07), the average number of clusters for design verification and debug problems (cid12–cid14, cid16) is consistently higher across all three models—Llama 3.1 405B, Claude 3.7 Sonnet, and GPT 4.1 as shown in Figure 2a. In the design verification categories, in addition to syntax and functional errors, failure entities include issues like "Misplaced SVA" and "Insufficient Coverage." To illustrate the diversity of failure types within design verification problems, we present a cluster visualization plot for Claude 3.7 Sonnet on Testbench Checker generat (cid13) using the PaCMAP graph reduction method (Wang et al. (2021)) in Figure 2b, which preserves both local and global distances.

| Cat. | Model Name | Pass Rate (%) | Category-Level Failure Analysis | | | |
| --- | --- | --- | --- | --- | --- | --- |
| | | | # Failed | # Clusters | Failed Entity of Max Cluster Size | Max Cluster Size (%) |
| cid02 | Llama 3.1 405B | 28.43% | 73 | 2 | Arbiter meltdown;Metastability hazards | 90.41% |
| | Claude 3.7 Sonnet | 42.16% | 59 | 2 | Data misalignment;Syntax errors | 55.93% |
| | GPT 4.1 | 37.25% | 64 | 10 | Encoding failures;Timing violations | 18.75% |
| cid03 | Llama 3.1 405B | 29.29% | 70 | 2 | Missing functionality | 57.14% |
| | Claude 3.7 Sonnet | 48.48% | 51 | 2 | Clock Domain;Protocol Violations | 54.90% |
| | GPT 4.1 | 39.39% | 60 | 3 | Reversed indexing; Module mismatch | 53.33% |
| cid04 | Llama 3.1 405B | 34.26% | 71 | 2 | Protocol Handling;Datapath Logic | 52.11% |
| | Claude 3.7 Sonnet | 45.37% | 59 | 2 | bit-slicing errors; missing states | 59.32% |
| | GPT 4.1 | 37.96% | 67 | 2 | Parameter Mismatch;Architecture Deviation | 52.24% |
| cid07 | Llama 3.1 405B | 17.31% | 86 | 3 | Logical Errors;Incomplete Implementation | 40.70% |
| | Claude 3.7 Sonnet | 36.54% | 66 | 2 | Structural Breakage;Area Shortfall | 54.55% |
| | GPT 4.1 | 23.08% | 80 | 2 | New mismatches;Unrequested signals | 56.25% |
| cid12 | Llama 3.1 405B | 20.00% | 80 | 3 | Missing coverage;Incorrect naming | 52.50% |
| | Claude 3.7 Sonnet | 25.00% | 75 | 4 | Missing timescale;Module mismatch | 56.00% |
| | GPT 4.1 | 12.00% | 88 | 3 | Truncated Implementation;Missing Tasks | 69.32% |
| cid13 | Llama 3.1 405B | 9.90% | 91 | 4 | Incorrect synchronization;Insufficient coverage | 35.16% |
| | Claude 3.7 Sonnet | 22.77% | 78 | 6 | Syntax errors;Unmatched blocks | 26.92% |
| | GPT 4.1 | 8.91% | 92 | 6 | Overhauled Testbench;Parameter Mismatch | 28.26% |
| cid14 | Llama 3.1 405B | 11.00% | 89 | 2 | Misplaced SVA;Operator Errors | 60.67% |
| | Claude 3.7 Sonnet | 25.00% | 75 | 2 | Flawed Timing;Syntax Mismatch | 58.67% |
| | GPT 4.1 | 13.00% | 87 | 2 | Procedural Blocks;Syntax Deviations | 58.62% |
| cid16 | Llama 3.1 405B | 34.65% | 66 | 2 | Datapath flaw;Protocol mismatch | 57.58% |
| | Claude 3.7 Sonnet | 58.42% | 42 | 9 | Faulty Reset Handling;Boundary Check Errors | 30.95% |
| | GPT 4.1 | 44.55% | 56 | 10 | Timer guard;Reset Logic | 23.21% |

Table 5: Failure analysis of Non-Agentic Generation, pass@1 ($n=1$). For each category, we show #failures, #clusters, top failure entity, and max cluster share (#failed cases of max cluster/#failed cases).

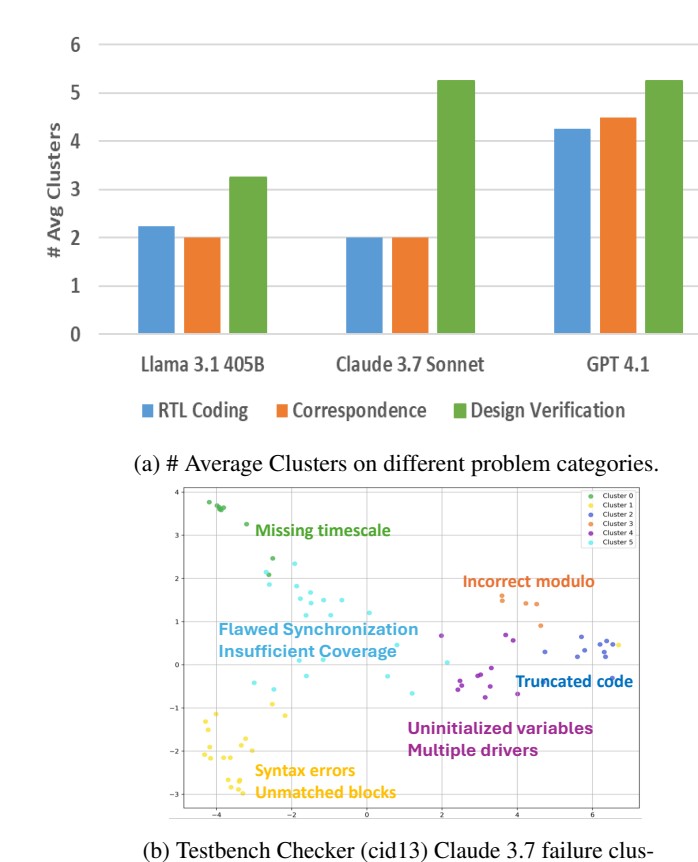

(a) # Average Clusters on different problem categories.

(b) Testbench Checker (cid13) Claude 3.7 failure cluster visualization.

Figure 2: Failure Analysis on different problem categories. Visualization plot uses PaCMAP graph reduction method (Wang et al. (2021)).

Lastly, we further analyze the Testbench Checker Generation set (cid13) after applying quality filtering (as shown in Table 1), since state-of-the-art LLMs achieve the lowest pass rates in this category, and a larger number of data points are filtered during the quality screening process among the design verification categories. Figure 3 presents the cluster visualizations of Llama 3.1 405B, Claude 3.7 Sonnet, and GPT-4.1 on design verification categories before and after quality filtering. Compared to the unfiltered data, the number of failure clusters is reduced after quality filtering due to decreased ambiguity and increased consistency in the problem descriptions. For Claude 3.7 Sonnet specifically, the number of failure clusters drops from 6 to 2 after quality filtering, reflecting the improved clarity of the case descriptions. In summary, our failure analysis reveals key challenges and insights into where state-of-the-art LLMs struggle across RTL tasks—particularly in design verification—offering valuable and comprehensive benchmarks for advancing LLM research in hardware design and verification.

## 6 LIMITATIONS

The XYZ benchmark is designed to push the limits of existing LLMs and agents in solving real-world hardware code generation tasks. While considerably more challenging for current large language models than prior benchmarks—particularly in areas such as design verification and module reuse—it does have limitations. The contexts of the Agentic datapoints are, on average, larger than those of the Non-Agentic datapoints. However, the Agentic context remains an oracle context and does not include files referencing additional units. The Question & Answer Code Comprehension datapoints do not sufficiently challenge the LLMs, and a separate task category focused on specification creation from RTL code may be more informative and demanding while addressing similar comprehension

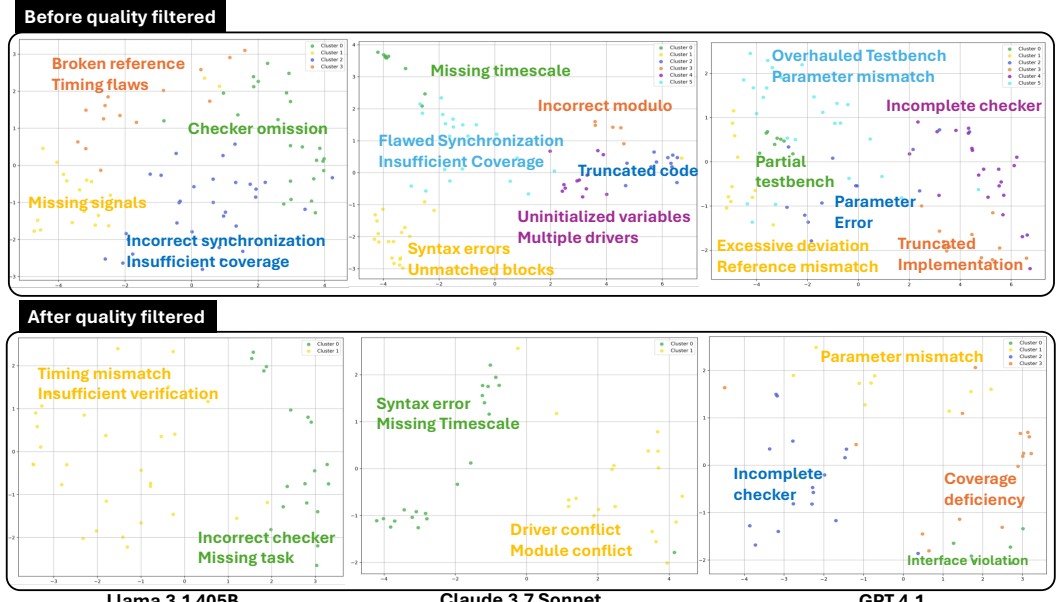

Figure 3: Failure cluster visualization of Testbench Checker set set (cid13) before/after quality filtered using the PaCMAP graph reduction method (Wang et al. (2021)). After quality filtered, the # of failure clusters is less because of improved ambiguity and consistency in prompt.

goals. Finally, the tasks in the benchmark are limited to standard hardware design and verification tasks and do not encompass the full range of challenges a design or verification engineer might face from project inception through fabrication. Specific academic and industry organizations may have additional requirements, custom tooling, or specialized needs not fully addressed by XYZ.

# 7 CONCLUSIONS

XYZ comprises 783 human expert-authored problems across 13 hardware design and verification task categories. The dataset spans Non-Agentic Code Generation, Non-Agentic Code Comprehension, and Agentic Code Generation tasks. State-of-the-art LLMs achieve no more than 34% pass@1 on Code Generation, revealing notable performance gaps—especially in design verification tasks such as SystemVerilog testbench generation. Given the tooling-intensive nature of hardware workflows, XYZ supports Dockerized agents and test harnesses for realistic tool interaction.

The Dockerized infrastructure not only enables sophisticated agent workflows, but also lowers the barrier to entry. Because the benchmark can be executed within portable container images, host system requirements are minimal and reproducibility across platforms is preserved. At the same time, the container-based approach is inherently extensible, allowing integration of additional commercial or open-source EDA tools, as well as future orchestration of full end-to-end flows.

While the current release focuses on common front-end design and verification tasks, semiconductor workflows span a much broader continuum that is often highly complex and institutionally specific. XYZ is designed with extensibility in mind, enabling incorporation of more advanced flows over time. For example, the infrastructure includes support for multiple tests per datapoint, yielding finer-grained diagnostic information about model performance and exposing more nuanced verification challenges.

Finally, the need for such infrastructure and datasets extends beyond the current benchmark. The long-term advancement of AI for semiconductor design and verification will depend on scalable and flexible evaluation environments that can evolve with the capabilities of both models and tools. By providing a rigorous yet adaptable foundation, XYZ aims to help drive this progress and catalyze continued research into AI-driven design flows.

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

## A  INFRASTRUCTURE DETAILS

The XYZ Benchmark implements a modular, containerized framework for evaluating hardware verification tasks, supporting both direct LLM evaluation and agent-based workflows. Its design emphasizes reproducibility, extensibility, and rigorous evaluation under diverse toolchains and environments.

The benchmark offers three complementary entry points that constitute the primary interface. The main execution utility functions as a unified evaluation engine for both LLMs and agents, supporting problem selection, model specification, and result collection. A companion utility extends this functionality to statistical evaluation by executing repeated trials and computing reliability estimates such as pass@k. A third utility generates structured reports from evaluation logs, providing both single-run analysis and aggregated statistical summaries. Together, these utilities provide a full workflow for executing, analyzing, and disseminating benchmark results. Configuration is entirely environment-based, with layered support for default settings, environment variables, and overrides, thereby enabling flexible deployment.

Two distinct evaluation paradigms are supported. In the non-agentic mode, language models are integrated directly through API calls. The system manages prompt preparation, response collection, and automated verification through containerized harnesses, enabling systematic comparison across models. The agentic mode instead relies on user-defined containers that are mounted with full problem contexts and toolchains, supporting iterative reasoning and tool use characteristic of agent workflows. This dual structure ensures that both conventional and experimental methodologies can be accommodated within a single framework.

Container orchestration is achieved through Docker Compose, which generates task-specific configurations to isolate agent execution from test harness verification. Agent containers are constructed around base images that encapsulate open-source hardware development environments, while verification harnesses rely on parallel configurations to ensure reproducible testing conditions. Two standardized base images serve as building blocks: a verification image containing open-source simulators (e.g., Icarus Verilog, Verilator) and an implementation image that includes Yosys for gate-level synthesis challenges. These images are used as stable reference environments, ensuring consistency across evaluations while allowing researchers to layer custom dependencies as needed. For commercial evaluation scenarios, user-provided base images integrate enterprise EDA tools such as Cadence Xcelium, with infrastructure support for license server connectivity and validation. Researchers are expected to extend these base environments when developing custom agents, thereby retaining compatibility with the verification pipeline while enabling specialized tool use.

Robust resource management ensures reproducibility even under constrained conditions. The system monitors workspace directories to guard against uncontrolled file growth, applies configurable timeouts to prevent indefinite execution, and automatically provisions isolated Docker networks for evaluation runs. Network policies currently provide container-level separation and controlled connectivity for commercial tool licensing.

Datasets are distributed in two complementary formats. A structured JSONL schema supports direct LLM evaluation by defining prompts, context, expected outputs, and verification procedures. An agentic schema expands these definitions into multi-file workspaces, enabling complex tool use and iterative reasoning strategies. Automatic transformation utilities allow researchers to convert between the two schemas while preserving semantic equivalence, ensuring that datasets can be reused across paradigms.

Evaluation metrics combine objective and subjective components. Objective verification is provided by the containerized test harnesses, yielding pass/fail results grounded in hardware development practice. Subjective scoring complements this by assessing explanation and comprehension tasks. Statistical extensions such as pass@k provide reliability estimates over repeated runs, accounting for the stochastic behavior of both LLMs and agents. Category-specific evaluation protocols further tailor metrics to the demands of code generation, comprehension, or design verification with commercial EDA tools.

Extensibility is a central feature. New models can be integrated through lightweight adapter files that register them with the evaluation framework, while local inference can be supported via standardized export/import routines that decouple prompt preparation from response evaluation. Agent develop-

ment is facilitated by containerized environments derived from Docker base images, allowing both open-source and commercial toolchains. To support researchers, the framework includes development templates and build scripts that provide practical starting points. These are intended as aids to custom agent creation rather than as reference implementations.

Finally, the framework has been engineered for scalability. Evaluation can be executed sequentially or in parallel, with automatic cleanup and resource monitoring to ensure stability under concurrency and low model accuracy. Deployment is supported with low host requirements, and with an environment-driven configuration system. This uniformity allows evaluations to be reproduced across diverse computational contexts with minimal modification.

# B COMPUTE REQUIREMENTS

Benchmark infrastructure development and model evaluation were performed on a virtual machine with 12 virtual CPUs and 24 GB of RAM, running Rocky Linux 8.10. Disk usage per model evaluation ranged from 6.4 GB to 15 GB, primarily due to errant RTL or testbench outputs generating large simulation logs or Verilog Change Dumps (VCDs). A built-in disk monitor in the XYZ framework checks each active datapoint run directory every second and aborts execution if its size exceeds 100 MB. Agents or models producing excessive output may trigger this limit. The framework also supports run directory compression.

Models were evaluated via API endpoints and are not included in the compute resource figures. Token usage per category can be estimated from Table 1.

# C FAILURE ANALYSIS

Section 5 presented a category-level analysis; here, we examine two specific examples to better understand where LLMs fail in generating correct code.

**Input Prompt**

Complete the existing `sorting_engine` module given below to implement the **brick sort** algorithm using finite state machine (FSM).

*[Brick sort description, algorithm example, and port list are omitted due to space constraints]*

**Parameters**
- `N` (Default is 8, Greater than 0): Number of elements to sort. Assume `N` is an even integer
- `WIDTH` (Default is 8, Greater than 0): Bit-width of each input element

**Latency Considerations**
Total latency = (N * (N - 1)) / 2 + 4
Perform a single compare-and-swap operation per clock cycle (sequential approach):
- **1 clock cycle** for moving from `IDLE` state to `LOAD`.
- **1 clock cycle** to load the data.

**Latency Considerations (Continued)**
- Perform **N passes** to completely sort the array.
  - Each **even-numbered pass** has N/2 comparisons and swaps.
  - Each **odd-numbered pass** has N/2-1 comparisons and swaps.
- Each comparison-and-swap takes **1 clock cycle**.
- **1 clock cycle** to transition to `DONE` state from the `SORT`.
- **1 clock cycle** to set the output sorted array and assert the `done` signal after sorting is complete.

**Latency Example:**
- N = 4, WIDTH = 4
- in_data = [0, 1, 2, 3]
- out_data = [3, 2, 1, 0]
- Latency = 10 clock cycles

**Golden Solution**

*[Only the core implementation is shown below due to space constraints]*

```
// Compare-swap the current pair
// Check if we are within the valid pair range
if (pair_idx < pairs_in_this_pass) begin
    if (pass_cnt[0] == 1'b0) begin
        if (data_array[2*pair_idx] > data_array[2*pair_idx+1]) begin
            {data_array[2*pair_idx], data_array[2*pair_idx+1]}
                <= {data_array[2*pair_idx+1], data_array[2*pair_idx]};
        end
    end
    else begin
        if ((2*pair_idx+2) < N) begin
            if (data_array[2*pair_idx+1] > data_array[2*pair_idx+2]) begin
                {data_array[2*pair_idx+1], data_array[2*pair_idx+2]}
                    <= {data_array[2*pair_idx+2], data_array[2*pair_idx+1]};
            end
        end
    end
end

// Update 'pass_cnt' in the same cycle as the last compare-swap to avoid a one-cycle delay
if (pair_idx + 1 == pairs_in_this_pass) begin
    pass_cnt <= pass_cnt + 1;
    pair_idx <= 0;
end else begin
    pair_idx <= pair_idx + 1;
end
```

**LLM-Generated Incorrect Solution**

*[Only the core implementation is shown below due to space constraints]*

```
if (pair_idx < pairs_in_this_pass) begin
    if (pass_cnt[0] == 1'b0) begin
        if (data_array[pair_idx*2] > data_array[pair_idx*2+1]) begin
            // Issue 1: Mixed usage of blocking '=' and non-blocking '<=' assignments
            temp = data_array[pair_idx*2];
            data_array[pair_idx*2] <= data_array[pair_idx*2+1];
            data_array[pair_idx*2+1] <= temp;
        end
    end else begin
        // Issue 2: missing check to ensure 'pair_idx*2+2' is within the valid range
        if (data_array[pair_idx*2+1] > data_array[pair_idx*2+2]) begin
            temp = data_array[pair_idx*2+1];
            data_array[pair_idx*2+1] <= data_array[pair_idx*2+2];
            data_array[pair_idx*2+2] <= temp;
        end
    end
    pair_idx <= pair_idx + 1;
end else begin
    // Issue 3: Updating 'pass_cnt' when pair_idx == pairs_in_this_pass
    // delays the update by one cycle after the last swap operation due to '<=' assignment,
    // causing a 1-cycle delay per pass,
    // which accumulates to an N-cycle delay for N elements to sort
    pass_cnt <= pass_cnt + 1;
    pair_idx <= 0;
end
```

Figure 4: A failure case on brick sort algorithm implementation.

## C.1 CASE STUDY 1:

Figure 4 highlights three critical flaws in the LLM-generated implementation of the brick sort algorithm, despite the fact that this algorithm is generally well understood by leading language models. First, the model carelessly mixes blocking (=) and non-blocking (<=) assignments, which can result in unintended behaviors due to mismatched update semantics. Second, it fails to perform bounds checking before accessing `data_array[2*pair_idx+2]`, potentially leading to out-of-range access. Most notably, the model delays updating the `pass_cnt` signal by one cycle after the final compare-and-swap in each pass, causing an extra cycle of latency per pass. Since brick sort performs exactly $N$ passes for an input of size $N$, this leads to a total of $N$ additional clock cycles, which violates the expected latency specified in the prompt.

These issues underscore a broader limitation of even the most capable LLMs: while they can reproduce high-level algorithmic structure, they often fail to account for cycle-accurate control sequencing, boundary conditions, and precise timing contracts critical for correct RTL behavior. The resulting code may appear syntactically correct, yet lacks the semantic fidelity expected in hardware design. This case study demonstrates that, despite recent advances, LLMs still fall short in generating accurate RTL code.

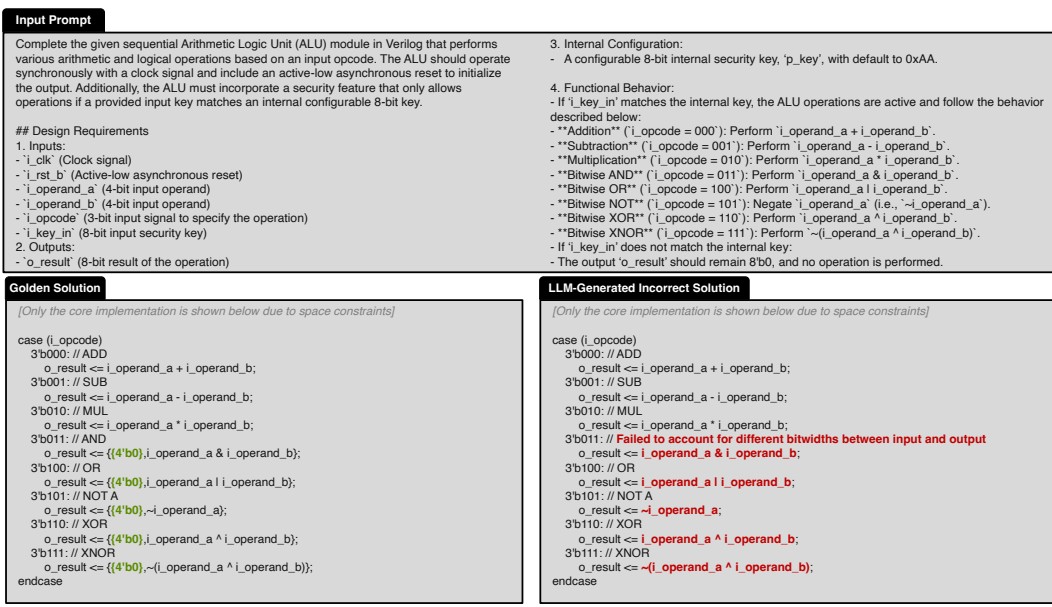

Figure 5: A failure case on ALU implementation.

## C.2 CASE STUDY 2:

Apart from their difficulty in reasoning about timing behavior, Figure 5 reveals a second critical limitation of LLMs for RTL coding: a tendency to ignore explicit bit-width handling. The incorrect implementation overlooks the width mismatch between the 4-bit operands (*i_operand_a*, *i_operand_b*) and the 8-bit output (*o_result*) for all bit-wise operations (AND, OR, NOT, XOR, XNOR). In RTL coding, assigning a 4-bit expression to an 8-bit target triggers an implicit zero-extension of the most significant bits. While this compiles, it silently violates the intent of specification: the upper four bits should be explicitly cleared so that downstream logic can rely on deterministic, intentionally driven zeros. The golden solution makes that intent explicit with {{4'b0}, ... } concatenations.

## D QUALITY FILTERING

Automatic quality filtering proceeds in two stages. The first stage applies sanity checks to the test harness: it must pass with the reference solution and fail with the initial context. The former ensures consistency between the test harness and the reference solution, while the latter confirms that a correct

solution is required and the initial context does not already satisfy the task. Of the 1,313 initial datapoints, 78 were excluded due to failing these sanity checks.

The second stage of quality filtering uses LLM-based judging with four metrics: ambiguity, consistency, category match, and behavioral match (Figure 6). The prompt used for this LLM quality judge is shown in Listing 1. It also includes fields for prompt refinement, enabling automated revisions of ambiguous or incorrect prompts; however, we do not report results from that experiment in this work, as further vetting is needed to ensure such revisions do not result in overly descriptive or trivial prompts. When running XYZ in map mode with the LLM judge, the output is a scored JSONL file with additional metadata fields. Post-processing scripts then combine the four metrics into an aggregate score and remove low-scoring problems. For this work, we used a threshold of 8.0 for a passing score. The final filtered JSONL dataset excludes the scoring fields.

The number of problems filtered per category is shown in Table 6, an expanded version of Table 1. Code Improvement (cid07) and Debugging (cid16) saw the most filtering, while Code Completion (cid02) saw the least. Pass rate changes resulting from quality filtering are shown in Table 7. Claude 3.7 showed a 10–14 point increase in pass rate for Code Improvement and Debugging, as did Spec-to-RTL (cid03) and Code Modification (cid04). Interestingly, Testbench and Assertion Generation (cid12–14) saw little improvement despite aggressive filtering, as shown in Table 6. This supports our findings in Section 4 that existing LLMs struggle fundamentally with generating correct and accurate SystemVerilog testbench code.

```
You are an expert at refining code challenge datapoints.
Analyze the provided datapoint and improve it, focusing ONLY on enhancing the 'prompt' field.
Improvements should be subtle and nuanced, and should not change the overall meaning of the datapoint,
but should make the datapoint more precise and helpful in solving the problem.
The 'input', 'output', 'categories', and 'harness' fields MUST remain unchanged and are provided
only for reference to help you understand the task better. Return a complete datapoint JSON structure
with an additional 'reasoning_prompt' field that explains your improvements, along with 'ambiguity_score'
and 'consistency_score' fields that evaluate the quality of the original datapoint.

You should also include a 'category_match_score' field that evaluates how well the category tag in the
original datapoint matches the category tag in the 'categories' field, where 1 means no match and 10
means a perfect match. With this, include a 'reasoning_category_match' field that explains your
reasoning for the category match score.

Additionally, provide a 'behavioral_match_score' field that evaluates how well the logic and behavior
    described in the prompt matches the actual logic and behavior in the output reference solution and what
    is checked in the test harness. Include a 'behavioral_match_reasoning' field explaining your assessment
    of this behavioral alignment.

I need help refining this code challenge datapoint (ID: {id}).

Here is the original datapoint:
```json
{json.dumps(datapoint, indent=2)}
```

IMPORTANT CONSTRAINTS:
- You can ONLY modify the 'prompt' field
- The following fields MUST remain exactly as they are in the original:
  * 'input': The input to the code challenge
  * 'output': The expected output of the code challenge (the "reference solution")
  * 'categories': Includes the difficulty ("easy", "medium", "hard") and the category tag. The category tags
      below are the only ones that are allowed:
    * 'cid002': "RTL Code Completion: Input must be skeleton code, and output must be the complete RTL code."
    * 'cid003': "Specification to RTL Translation: Input must be a natural language specification, and output
        must be the complete RTL code."
    * 'cid004': "RTL Code Modification: Input must be existing RTL code and natural language specification of
        the changes to make, and output must be the modified RTL code."
    * 'cid005': "Specification to RTL Translation - Module Instantiation and Component Reuse: Input must be a
        natural language specification, and output must be the complete RTL code with module instantiations
        and component reuse."
    * 'cid006': "RTL Correspondence (Match RTL to Specification or vice versa): Input must be an RTL code and
        a natural language specification, and output must be the RTL code that matches the specification, or
        vice-versa."
    * 'cid007': "RTL Lint Improvement or Power-Performance Optimization: Input must be an RTL code and a
        natural language specification of the changes to make, and output must be the linted or optimized
        RTL code. For power-performance optimization, the specification should clearly specify criteria of
        area redunction or latency changes."
    * 'cid008': "Testbench Correspondence (Match Testbench to Test Plan or vice versa): Input must be a
        testbench and a test plan, and output must be the testbench that matches the test plan, or vice-
        versa."
    * 'cid009': "Question & Answer on RTL: Input must be an RTL code and a question, and output must be the
        answer to the question based on the RTL code."
    * 'cid010': "Question & Answer on Testbench: Input must be a testbench and a question, and output must be
        the answer to the question based on the testbench."
    * 'cid012': "Test Plan to Testbench Stimulus Generation: Input must be a test plan, and output must be the
        stimulus for the testbench without any logic to check the output of the device under test."
    * 'cid013': "Test Plan to Testbench Checker Generation: Input must be a test plan, and output must be the
        checker for the testbench that can be used to verify the output of the device under test along with
        stimulus generation. The input might also include an existing stimulus-only testbench, in which case
```

```
                    the output should be a checker that can be used to verify the output of the device under test along
                    with the existing stimulus."
              * 'cid014': "Test Plan to Assertions Generation: **Must** be about generating assertions for the testbench
                    . The input will include a test plan and existing testbench, and the output must include the
                    assertions for the testbench."
              * 'cid016': "RTL Debugging and Bug Fixing: **Must** be about fixing an existing bug in the RTL that is
                    leading to incorrect output. The input will include an RTL code and a testbench, and the output must
                    include the fixed RTL code."
          * 'harness': The harness that the code challenge uses to eveluate the output
    - These fields are provided only as reference to help you understand the task
    - You don't need to include these unchanged fields in your response - only include 'prompt', 'reasoning_prompt
        ', and the score fields
    - When in doubt, be more critical of the datapoint and give lower scores. Critical information may be missing
        from the datapoint, or there may be a bug in the harness and reference solution in matcing a
        specification in the prompt.
    - The person who will be using the refined datapoint **will not be granted access** to the reference solution
        or harness, so they must rely on the datapoint (prompt, input, context, etc.) and their own knowledge to
        make the best possible solution. Therefore, refrain from referring to the src/ directory in any prompt
        revisions.

    Please provide a refined version of the datapoint that:
    1. Clarifies and enhances ONLY the 'prompt' field
    2. Makes the instructions more precise and helpful based on examining the input, output, and test harness
    3. Adds hints or clarifications that would help a senior hardware engineer succeed
    4. Should not solve the problem in the refined prompt - only add hints or critical clarifications that would
        not be assumed by a senior hardware engineer
    5. Maintains the exact same structure for all other fields (if you include them)
    6. Adds a 'reasoning_prompt' field explaining your improvements and why they help
    7. Includes an 'ambiguity_score' rating from 1-10 for the original prompt (1 = very vague/impossible to solve,
        10 = perfectly clear)
    8. Includes a 'consistency_score' rating from 1-10 for how well the original problem components align (1 =
        inconsistent between prompt/input/output/harness, 10 = perfectly consistent)
    9. Includes a 'category_match_score' rating from 1-10 for how well the category tag in the original datapoint
        matches the category tag in the 'categories' field (1 = there is a better category for the datapoint, 10
        = perfect match)
    10. Includes a 'behavioral_match_score' rating from 1-10 that evaluates how well the logic and behavior
        described in the prompt matches the actual logic and behavior in the output reference solution and what
        is checked in the test harness (1 = significant mismatch, 10 = perfect behavioral alignment)

    The 'reasoning_prompt' field should contain your justification for the prompt improvements you made, what
        issues you addressed, and how these enhancements will help the model succeed. Your reasoning should also
        address the three scores (ambiguity, consistency, and category match) in your explanation.

    The 'ambiguity_score' should reflect how clear or ambiguous the original prompt was, where 1 means extremely
        vague/impossible to understand and 10 means completely clear with no ambiguity. Ambiguity is a measure
        of how well a senior hardware engineer would be able to understand the prompt and solve the problem
        without having to iterate multiple times.

    The 'reasoning_ambiguity' field should explain your reasoning for the ambiguity score.

    The 'consistency_score' should reflect how well the various components of the problem (prompt, input, output,
        harness) align with each other, where 1 means severe inconsistencies and 10 means perfect alignment. In
        particular, the prompt should match with the reference solution ('output') and the harness very closely.

    The 'reasoning_consistency' field should explain your reasoning for the consistency score.

    The 'category_match_score' should reflect how well the category tag in the original datapoint matches the
        category tag in the 'categories' field. When scoring, consider if the task better fits in a different
        category.

    The 'reasoning_category_match' field should explain your reasoning for the category match score.

    The 'behavioral_match_score' should evaluate specifically how well the logic and behavior described in the
        prompt matches the actual implementation details in the reference solution and what is being checked in
        the test harness. It focuses on the technical alignment of the expected behavior versus what is actually
        implemented and tested.

    The 'behavioral_match_reasoning' field should explain your reasoning for the behavioral_match_score,
        highlighting any discrepancies or strong alignments between the prompt's behavioral specifications and
        the actual implementation/testing.

    Your JSON response can be minimal, containing just:
    ```json
    {
      "prompt": "your improved prompt here",
      "reasoning_prompt": "your explanation here",
      "ambiguity_score": 8,
      "reasoning_ambiguity": "your explanation for ambiguity_score here",
      "consistency_score": 8,
      "reasoning_consistency": "your explanation for consistency_score here",
      "category_match_score": 8,
      "reasoning_category_match": "your explanation for category_match_score here",
      "behavioral_match_score": 8,
      "behavioral_match_reasoning": "your explanation for behavioral_match_score here"
    }
    ```

    Or you can include the full datapoint structure if you prefer. The system will ensure other fields remain
        unchanged.

    Return the datapoint as valid JSON.
```

Listing 1: Quality filtering prompt instructions.

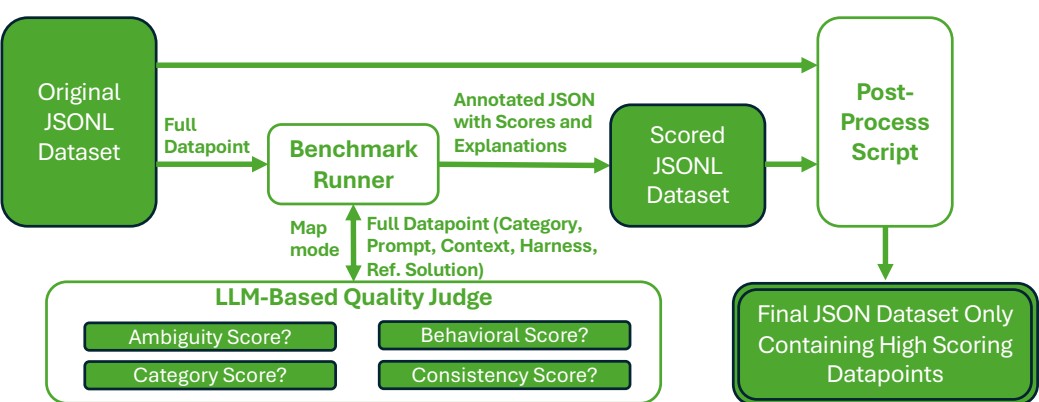

Figure 6: Quality filtering flow.

| Type | ID | Category Description | % Filtered | Unfiltered Volume | | Filtered Volume | |
|---|---|---|---|---|---|---|---|
| | | | | Non-Agnt | Agentic | Non-Agnt | Agentic |
| **Code Generation** | **cid02** | RTL – Code Completion | 7.8% | 102 | 0 | 94 | 0 |
| | **cid03** | RTL – Natural Language Spec to Code | 26.8% | 99 | 58 | 78 | 37 |
| | **cid04** | RTL – Code Modification | 46.4% | 108 | 45 | 56 | 26 |
| | **cid05** | RTL – Spec to RTL (Module Reuse) | 35.0% | 0 | 40 | 0 | 26 |
| | **cid07** | RTL – Code Improvement (Linting/QoR) | 60.6% | 104 | 0 | 41 | 0 |
| | **cid12** | Design Verification – Testbench Stimulus | 35.8% | 100 | 34 | 68 | 18 |
| | **cid13** | Design Verification – Testbench Checker | 45.0% | 101 | 28 | 53 | 18 |
| | **cid14** | Design Verification – Assertion Generation | 31.5% | 100 | 43 | 68 | 30 |
| | **cid16** | Design Verification – Debugging / Bug Fix | 64.1% | 101 | 30 | 36 | 11 |
| **Code Comprehension** | **cid06** | Correspondence – RTL to/from Spec | 43.3% | 60 | 0 | 34 | 0 |
| | **cid08** | Correspondence – Testbench to/from Plan | 47.3% | 55 | 0 | 29 | 0 |
| | **cid09** | Question & Answer – RTL | 38.2% | 55 | 0 | 34 | 0 |
| | **cid10** | Question & Answer – Testbench | 48.0% | 50 | 0 | 26 | 0 |
| | | **Total # of Problems** | **40.4%** | **1035** | **278** | **617** | **166** |

Table 6: Comparison of Non-Agentic and Agentic problem counts by task category, with percentage of problems removed by filtering.

| Model | cid02 | cid03 | cid04 | cid05 | cid07 | cid12 | cid13 | cid14 | cid16 |
|---|---|---|---|---|---|---|---|---|---|
| **Claude 3.7 Sonnet** | 0.35 | 14.96 | 15.62 | -0.15 | 14.2 | 6.88 | -0.03 | 2.13 | 10.88 |
| " Thinking | 0.38 | 9.95 | 15.86 | 1.35 | 14.79 | 6.69 | 1.45 | 4.23 | 17.75 |
| **Claude 3.5 Haiku** | 1.2 | 7.37 | 5.44 | -2.23 | 10.11 | 3.05 | -0.53 | 3.77 | 11.07 |
| **GPT 4.1** | 0.52 | 7.52 | 9.23 | -3.23 | 10.75 | 3.61 | 6.63 | 1.54 | 4.03 |
| **GPT o1** | 0.23 | 5.11 | 1.61 | 0.69 | 8.41 | 4.6 | 2.94 | 1.37 | 17.14 |
| **GPT o4-mini** | 0.56 | 8.85 | 9.91 | -0.5 | 9.24 | 2.59 | 1.94 | 2.34 | 12.17 |
| **Llama 3.1 405B** | 0.31 | 9.15 | 7.81 | 6.38 | 5.19 | 1.91 | -2.69 | 2.69 | 17.83 |
| **Llama 3.1 70B** | -0.38 | 6.13 | 4.87 | 4.19 | 7.71 | -0.8 | 2 | -1.16 | 3.03 |

Table 7: Change in pass@1 ($n = 5$) rates after quality filtering for Code Generation tasks. Positive values indicate improved pass rates. Units are percentage points.

| Model | cid006 | cid008 | cid009 | cid010 |
|---|---|---|---|---|
| **Claude 3.7 Sonnet** | 0.07 | 0.04 | 0.11 | 0.10 |
| **" Thinking** | 0.09 | 0.10 | 0.08 | 0.11 |
| **Claude 3.5 Haiku** | 0.03 | -0.01 | 0.13 | 0.12 |
| **GPT 4.1** | 0.04 | 0.03 | 0.12 | 0.09 |
| **GPT o1** | 0.03 | -0.01 | 0.11 | 0.08 |
| **GPT o4-mini** | 0.02 | 0.02 | 0.07 | 0.08 |
| **Llama 3.1 405B** | 0.04 | 0.01 | 0.13 | 0.10 |
| **Llama 3.1 70B** | 0.04 | 0.00 | 0.13 | 0.10 |

Table 8: Change in average score ($n = 5$) after quality filtering for Code Comprehension tasks. Positive values indicate improved performance. Units represent the difference in average score.

# E    SUPPLEMENTAL: REPRODUCIBILITY

We recognize the central importance of reproducibility for both validating prior work and enabling future advances. As this work introduces an evaluation benchmark, it is especially critical that the community can reliably reproduce our reported results and then apply the same infrastructure for consistent comparisons in future research. To this end, our benchmark infrastructure and dataset are fully released under permissive open-source licenses and are publicly accessible on GitHub and Hugging Face. The complete infrastructure—including Docker scaffolding, open-source EDA tool images, and evaluation data points—allows independent researchers to validate our results directly, while also serving as a shared foundation for future benchmarking studies. All released artifacts are versioned, openly licensed, and designed for long-term accessibility by the community.

In accordance with ICLR's double-blind review policy, we cannot provide direct repository links within the submission. However, we confirm that these resources are publicly available today, and we would be happy to provide the links to the track chair to verify their accessibility and reproducibility. Community resources are also in place to continue maintaining and supporting the benchmark over time.

