# OpenReview forum: "Can LLMs Design Real Hardware? A New Benchmark for RTL Design and Verification Tasks"
_ICLR.cc/2026/Conference — Submitted to ICLR 2026_

### Official Review · Reviewer_hfww · 2025-10-27

**Soundness:** 2
**Presentation:** 1
**Contribution:** 2
**Rating:** 4
**Confidence:** 3

**Summary:**

This paper proposes a dataset containing 783 challenging problems across 13 tasks, aiming to broaden the problem scope and increase the complexity of development. This work evaluates several LLM backbones on specific tasks and analyzes model failures.

**Strengths:**

- The work is good in problem coverage and difficulty, which brings a broader distribution of problems orthogonal to existing benchmarks.
- The authors present model failures and provide observations on why they fail.

**Weaknesses:**

- Each category has only dozens of problems in the dataset due to the fine-grained splitting.
- The writing needs improvement. For instance, Contribution 4 is quite lengthy, and I had to read it several times to understand it fully. Also, there are awkward line breaks in the titles in Table 1. Additionally, could you clarify whether “ Thinking refers to Claude 3.7 Sonnet with or without the extended thinking feature?
- There is no need to use such a large figure for Figure 2, as it contains little information. Space could be used to present visualizations of specific problems currently lacking in the main text.
- Since the dataset is currently not available, I cannot fairly judge whether the problems share good quality and standard.

**Questions:**

See weaknesses.

---

### Official Review · Reviewer_umVY · 2025-10-29

**Soundness:** 3
**Presentation:** 2
**Contribution:** 2
**Rating:** 4
**Confidence:** 4

**Summary:**

This work presents a comprehensive benchmark and supporting infrastructure for RTL design and verification. The benchmark includes 783 problems across 13 task categories, covering RTL generation, verification, debugging, specification alignment, and technical Q&A, all curated by experienced hardware engineers. By incorporating both non-agentic and agentic tasks, it establishes more realistic and challenging evaluation settings than previous benchmarks. Under these conditions, state-of-the-art models achieve no more than 34% pass@1 accuracy on code generation tasks, with agentic tasks involving RTL reuse and verification proving particularly difficult. A detailed failure analysis further investigates the causes of these failures and highlights the key domain-specific capabilities that LLMs must develop to improve. Overall, the benchmark reveals substantial limitations in current model performance, emphasizing the need for continued progress toward robust and practical hardware design automation.

**Strengths:**

The paper’s main strength lies in its significance: it introduces a challenging and comprehensive benchmark for RTL design and verification that pushes the limits of current LLMs and agents, addressing a critical gap in realistic hardware design evaluation. By incorporating both non-agentic and agentic settings, it aligns with emerging trends in agentic AI and provides a strong foundation for studying autonomous design agents in practical workflows. The benchmark is methodologically solid, featuring carefully curated, expert-authored problems, a reproducible Dockerized infrastructure, and a rigorous evaluation protocol. Moreover, the detailed failure analysis offers valuable insights into where current models fall short, helping to guide future research. Overall, the work is technically robust and impactful, providing a timely and important resource for advancing LLM-driven hardware design research.

**Weaknesses:**

While the paper presents a valuable benchmark, its technical contribution is relatively limited, as it primarily focuses on dataset construction and evaluation rather than introducing new algorithms or modeling techniques. The current evaluation is also not comprehensive enough for a benchmark paper: it only tests a small subset of popular LLMs (Claude, GPT, and Llama), whereas broader coverage, including models like Gemini, Qwen, DeepSeek, and domain-specific models such as RTLCoder[1], CodeV[2], and CraftRTL[3], would provide a more meaningful reference for the community. Moreover, all agentic problems are evaluated in single-turn (non-agentic) mode, which undermines one of the benchmark’s core claims of supporting agentic evaluation. Furthermore, the evaluation reports only pass@1 accuracy, omitting standard metrics such as pass@5 and pass@10 that are commonly used in code generation benchmarks to reflect sampling variability. In addition, the paper lacks experiments exploring reasoning-augmented inference methods, such as chain-of-thought prompting or tool-assisted reasoning, which would better assess model robustness in realistic agentic settings. Finally, the paper contains several formatting and compliance issues: table titles appear below instead of above the tables, citations use inconsistent commands (*e.g.*, \citet instead of \citep), and the required sections on ethics, reproducibility, and LLM usage are missing. The writing style also shows clear signs of extensive LLM assistance, yet the authors do not acknowledge or document this usage.

[1] Liu, Shang, et al. "Rtlcoder: Fully open-source and efficient llm-assisted rtl code generation technique." *IEEE Transactions on Computer-Aided Design of Integrated Circuits and Systems* (2024).

[2] Zhao, Yang, et al. "Codev: Empowering llms with hdl generation through multi-level summarization." *IEEE Transactions on Computer-Aided Design of Integrated Circuits and Systems* (2025).

[3] Liu, Mingjie, et al. "CraftRTL: High-quality Synthetic Data Generation for Verilog Code Models with Correct-by-Construction Non-Textual Representations and Targeted Code Repair." *The Thirteenth International Conference on Learning Representations*.

**Questions:**

1. What is the specific criterion used to distinguish non-agentic from agentic problems in the benchmark? For instance, the benchmark does not include any agentic tasks for code improvement. However, in my perspective, this category inherently involves iterative LLM interactions and tool usage. Could you clarify the rationale behind this categorization?
2. Some tasks require commercial tools for execution, which could limit accessibility, particularly for academic users. Have you considered or could you propose alternative solutions to ensure broader usability?
3. Section 3 mentions a “map feature” supporting automated quality filtering. Could you clarify how this filtering is implemented—specifically, whether it occurs before evaluation or during runtime—and why cases that fail the filtering are not discarded directly?
4. For the correspondence tasks, the paper mentions that model outputs must match the reference responses verbatim. Could you explain why verbatim matching is required and how this task is evaluated in practice, especially when semantically equivalent but lexically different responses could occur?
5. The Q&A task uses GPT-o4-mini as the evaluation model. Given that this model also participates as a test subject, could this introduce self-evaluation bias that inflates its scores? Have you tested cross-evaluation or human calibration to assess this effect?
6. The failure analysis is based on *pass@1* with *n = 1*, which introduces substantial randomness and may reduce result reliability. Could you justify this choice or provide evidence that your findings are robust to sampling variation?

---

### Official Review · Reviewer_X8cT · 2025-10-31

**Soundness:** 2
**Presentation:** 1
**Contribution:** 3
**Rating:** 2
**Confidence:** 3

**Summary:**

This paper introduces XYZ, a benchmark designed to advance research in LLMs and AI agents for hardware design and verification. It comprises a wide range of expert-authored tasks in these domains. Compared to prior benchmarks, XYZ presents greater challenges and is the first to evaluate Dockerized LLM-based agents on real-world problems involving EDA tool usage. The study evaluates several state-of-the-art models using the proposed benchmark and provides a comprehensive analysis of the results. In particular, the paper conducts a detailed failure analysis and offers valuable insights to guide future LLM research in hardware design and verification. Limitations of the benchmark are also reported and discussed.

**Strengths:**

1. The benchmark introduced in this work is more challenging than those in previous studies, which helps reveal significant gaps in the capabilities of current models and may thereby stimulate further research toward robust, real-world hardware design automation.
2. It is the first benchmark capable of evaluating Dockerized LLM-based agents on real-world problems that require EDA tool usage, thereby addressing a critical gap in prior work.
3. This study evaluates several state-of-the-art models using the proposed benchmark and offers a comprehensive analysis of the results. Notably, it includes a detailed failure analysis and provides valuable insights intended to guide and inspire future research in this field.

**Weaknesses:**

1. **Poor Readability and Misleading Categorization**:

    The current taxonomy of tasks in the benchmark lacks clarity. For instance, placing “verification” under “code generation” is conceptually inconsistent. A reorganization of task categories—such as distinguishing between “code generation” and “code comprehension,” “agentic” and “non-agentic” tasks, or “verification” and “coding”—would improve clarity and logical grouping.

2. **Dependence on Commercial Tools**:

    The use of proprietary or commercial tools in the benchmark limits its reproducibility and accessibility. Not all researchers may have access to these tools, which restricts the ability to fully evaluate or compare methods in a standardized manner.

3. **Insufficient Emphasis on Benchmark Methodology**:

    While the paper presents extensive experimental results on various LLMs, it underemphasizes the benchmark’s design methodology. A benchmark should clearly justify its construction, demonstrate its advantages over existing alternatives, and provide a transparent evaluation protocol for future users.

4. **Overextension into Failure Analysis**:

    The failure analysis section, while interesting, falls outside the core scope of a benchmark paper. The focus should remain on the benchmark’s design, motivation, and a summary of baseline model performances—not detailed error diagnostics.

5. **Inadequate Discussion of Agent Evaluation**:

    Although a key contribution is the evaluation of agents (not just LLMs), the paper lacks a thorough description of the tools available to the agents, the robustness of tool integration, and the agents’ capabilities in using these tools. This limits the interpretability of agent-related results.

**Questions:**

1. **Validation of “Real Hardware” Claims**:

    The paper states that the benchmark is based on “real hardware” problems, but provides limited evidence to support this claim. It would be helpful to include metrics such as average code complexity or evidence of sourcing from real-world development projects—rather than solely referencing collaboration with engineers.

2. **Evaluation Metrics for Specific Task Types**:

    The criteria for evaluating performance across different task categories are unclear. For example, how is the “pass rate” defined and measured for “assertion generation”? Similarly, how is performance assessed for “RTL to/from specification” tasks? A clear explanation of the evaluation protocol for each category is needed.

3. **Inconsistencies in Task Categorization and Labeling**:

    In Table 1, the entry “cid01” appears to be missing. What does “cid” refer to? Additionally, categorizing “RTL to/from specification” under “code comprehension” seems inappropriate, as generating RTL from specifications is fundamentally a generation task. There also appears to be redundancy with “spec to RTL (cid05).”

4. **Comparison with Existing Benchmarks**:

    Several sub-domains covered in this benchmark (e.g., assertion generation) already have established benchmarks (e.g., FVEval). It would be valuable to clarify how this benchmark differs—what are its unique strengths and limitations? Given its broad scope, is it sufficiently deep in each specific area compared to specialized benchmarks?

---

### Official Review · Reviewer_Q3bn · 2025-10-31

**Soundness:** 3
**Presentation:** 3
**Contribution:** 3
**Rating:** 4
**Confidence:** 4

**Summary:**

The paper introduces XYZ benchmark for evaluating LLMs on hardware design tasks that include Verilog code generation tasks and verification tasks. The benchmark is oriented towards both agentic and non-agentic evaluations.

**Strengths:**

- The paper introduces a hard benchmark for evaluating LLMs on hardware design tasks. The benchmark is harder than current benchmarks such as VerilogEval and RTLLM and includes more broader categories including verification tasks.

**Weaknesses:**

- There is no mention of the problem scale/size. Lines of code or gate size don't necessarily correlate with hardness level, but still a view of how big or small are the problems could be helpful, especially that the title mentions real hardware which could imply SoC level problems.
- Agentic problems were converted to non-agentic format which leaves the study a bit incomplete. A simple baseline using an existing agentic framework would have been valuable to demonstrate how LLMs perform on this benchmark when operating in agentic mode.
- The paper relies on LLM as an automatic judge for comprehension tasks, but provides no validation experiments demonstrating the accuracy or reliability of its scoring. In particular, there is no analysis of its correlation with human evaluation, raising concerns about the credibility of the reported comprehension results.
- The paper doesn't evaluate LLM models finetuned specifically for Verilog, which raises the question of whether LLMs underperform on this benchmark due to dataset complexity or due to the general purpose nature of LLMs evaluated.

**Questions:**

1) The paper states that agentic problems were converted to non-agentic format for evaluation. Can the authors clarify why a baseline agentic framework (simple tool-calling loop) was not implemented?
2) Can authors provide analysis on gate level complexity of the benchmark problems ?

---

### Meta-Review · Area_Chair_errA · 2026-01-06

**Summary:**

This paper introduces a large-scale benchmark and evaluation infrastructure for assessing Large Language Models (LLMs) and LLM-based agents on hardware design and verification tasks.

**Reviewer Concerns:**

- Agentic Evaluation Is Incomplete or Undermined

- Task categorization is confusing or inconsistent

- The paper claims the benchmark reflects “real hardware,” but provides little quantitative evidence:
No clear reporting of lines of code, module size, or gate-level complexity
No concrete comparison to SoC- or IP-level designs
Reviewers wanted stronger evidence beyond anecdotal sourcing from engineers.

- Reliability of LLM-Based Evaluation

- Only a small set of general-purpose LLMs are evaluated.

- Dependence on commercial/proprietary EDA tools limits accessibility.

- Dataset and infrastructure were not available at review time.

- Presentation issues (formatting, table placement, writing quality) noted by multiple reviewers.

**Reviewer Scores:**

Reviewer Q3bn (Score: 4, borderline reject)

Reviewer X8ct (Score: 2, reject)

Reviewer umVY (Score: 4, borderline reject)

Reviewer hfww (Score: 4, borderline reject)

No rebuttal happened, score remains

---

### Decision · Program_Chairs · 2026-01-26

Reject